# Measurement Method Based on Multispectral Three-Dimensional Imaging for the Chlorophyll Contents of Greenhouse Tomato Plants

**DOI:** 10.3390/s19153345

**Published:** 2019-07-30

**Authors:** Guoxiang Sun, Xiaochan Wang, Ye Sun, Yongqian Ding, Wei Lu

**Affiliations:** 1College of Engineering, Nanjing Agricultural University, Nanjing 210031, China; 2Jiangsu Province Engineering Lab for Modern Facility Agriculture Technology & Equipment, Nanjing 210031, China

**Keywords:** multispectral, three-dimensional reconstruction, greenhouse tomato, chlorophyll, SPAD, plant phenotypes

## Abstract

Nondestructive plant growth measurement is essential for researching plant growth and health. A nondestructive measurement system to retrieve plant information includes the measurement of morphological and physiological information, but most systems use two independent measurement systems for the two types of characteristics. In this study, a highly integrated, multispectral, three-dimensional (3D) nondestructive measurement system for greenhouse tomato plants was designed. The system used a Kinect sensor, an SOC710 hyperspectral imager, an electric rotary table, and other components. A heterogeneous sensing image registration technique based on the Fourier transform was proposed, which was used to register the SOC710 multispectral reflectance in the Kinect depth image coordinate system. Furthermore, a 3D multiview RGB-D image-reconstruction method based on the pose estimation and self-calibration of the Kinect sensor was developed to reconstruct a multispectral 3D point cloud model of the tomato plant. An experiment was conducted to measure plant canopy chlorophyll and the relative chlorophyll content was measured by the soil and plant analyzer development (SPAD) measurement model based on a 3D multispectral point cloud model and a single-view point cloud model and its performance was compared and analyzed. The results revealed that the measurement model established by using the characteristic variables from the multiview point cloud model was superior to the one established using the variables from the single-view point cloud model. Therefore, the multispectral 3D reconstruction approach is able to reconstruct the plant multispectral 3D point cloud model, which optimizes the traditional two-dimensional image-based SPAD measurement method and can obtain a precise and efficient high-throughput measurement of plant chlorophyll.

## 1. Introduction

The growth, yield, and quality of crops depend on a variety of nutrient elements, among which nitrogen has the most significant impact [1,2]. Because 70–80% of nitrogen in the leaves is concentrated in the chloroplasts, the chlorophyll content reflects the nitrogen nutritional status of the plant [1,3,4]. The concentration of chlorophyll, a critical pigment related to photosynthesis, is often used as an important indicator for evaluating a plant’s photosynthesis ability, growth status, environmental stress, nutritional stress, and pest and disease stress [5,6,7]. Therefore, an efficient, accurate, and nondestructive method for measuring the chlorophyll content of plants is of great significance for the rapid assessment of plant growth status and, as a result, precise cultivation management and accurate plant yield estimation.

Spectrophotometry is the most commonly used technique for performing routine measurements of chlorophyll content in plant leaves [2,8]. In particular, the Arnon method is used throughout the world [8]; it requires a series of operational procedures, involving leaf grinding, filtration, and centrifugation, to remove debris. Although it yields accurate and reliable results, this method is destructive, costly, time-consuming, and complicated to implement because chlorophyll pigments are prone to photolytic decomposition. Hence, spectrophotometry is not an ideal tool for regular, systematic plant research because it cannot meet the requirements for high-throughput measurements and is unsuitable for field monitoring.

The rapid development of sensing and computer vision technologies has enabled the nondestructive assessment of plant chlorophyll using monocular vision [9,10,11,12,13], laser scanning imaging [14], multispectral imaging [15,16], hyperspectral imaging [17,18,19], chlorophyll fluorescence imaging [20], and other techniques, combined with image-processing methods. The spectral reflectance data in certain wavebands of light [2,6,7] can be used to establish a chlorophyll measurement model that rapidly and effectively predicts the chlorophyll contents of plants. However, the performance of prediction models that directly use spectral data in the relevant wavebands is prone to the effects of numerous factors, such as the soil, atmospheric, and environmental conditions, as well as the leaf structure. Therefore, a vegetation index, which is typically established by combining data from multiple relevant wavebands, is often used to construct a chlorophyll measurement model. This approach effectively reduces the impact of external factors and improves the prediction performance and stability of the model [2,6,7].

Monocular vision-based chlorophyll assessment technology primarily uses characteristic parameters or combinations of 2D images in RGB, HSV, YUV, or LAB color spaces [10,13] to establish a plant chlorophyll assessment model. This technology requires images that are taken under very high environmental illumination and its assessment model has relatively low applicability. Hyperspectral imaging is frequently applied to determine chlorophyll-sensitive wavebands using methods involving the correlation coefficient, inverse spectrum, spectral difference analysis, stepwise regression, discriminant analysis, partial least-squares regression, principal component analysis, continuous projection algorithms, independent component analysis, and artificial neural networks [2,6,7]. After the characteristic wavebands are selected, multispectral imaging is applied to obtain the actual chlorophyll measurement. The measurement model is constructed using a linear or nonlinear method according to the spectral reflectivity or vegetation indexes at the characteristic wavebands. The linear methods include principal component regression, linear regression, multiple linear regression, and partial least-squares regression. Nonlinear modeling methods include an artificial neural network, soft independent modeling of class analogies, a support vector machine, and the kernel partial least-squares method [2,6,7]. Chlorophyll fluorescence imaging technology provides physiological information, such as plant nutrients, diseases, and stresses, by using characteristic manifestations in plant fluorescence images [20]. This technology has strict requirements for the imaging environment, excitation light source, and imaging devices; hence, it cannot be widely promoted and applied because its operational conditions are limited and it requires very expensive equipment.

At present, all plant chlorophyll measurement technologies, including monocular vision and multispectral, hyperspectral, and chlorophyll fluorescence imaging technologies, extract the characteristic parameters from 2D images of plants (mostly plant leaves, seedling plants, or canopy images) to establish a chlorophyll measurement model. However, because plants have a complex 3D morphology, it is difficult to obtain accurate and stable results using these 2D-image-based technologies and to determine the 3D distribution of chlorophyll using only single-view plant canopy imaging data. To date, plant 3D reconstruction methods have been extensively developed on the basis of 3D LiDAR data [21,22], the Kinect camera [23,24], multiview photogrammetry [25,26], and multicamera synchronous reconstruction [27]. Kinect sensor-based 3D plant reconstruction can be divided into single-view [28] and multiview reconstruction [29], the latter mainly using the iterative closest point (ICP) algorithm [23,30]. However, first of all, we need to solve the problem around the rough registration of multiview point clouds, otherwise ICP cannot be used for accurate registration. Furthermore, a reconstructed plant 3D point cloud model contains only coordinate information and RGB color information. Various studies have been conducted to reconstruct multispectral 3D point cloud models. Using plant images obtained from 50 angles of view (AOVs), Kenta et al. [31] constructed a plant 3D point cloud model that was based on the “structure from motion” theorem and developed a chlorophyll measurement model using a normalized red value, r/(r + g + b). Using multispectral plant canopy images obtained from 24 AOVs, Zhang et al. [32] established a multispectral 3D point cloud model and a regression model using the difference vegetation index (DVI), normalized DVI (NDVI), green NDVI (GNDVI), and ratio vegetation index (RVI). The relative chlorophyll content was measured by a soil and plant analyzer development (SPAD). However, the construction of a multispectral 3D point cloud model is inefficient, because plant images from dozens of AOVs are required. Thus, this technology is unsuitable for high-throughput chlorophyll measurement.

In this study, depth vision technology and multispectral imaging technology were jointly applied to obtain images of the study subjects, namely, greenhouse tomato plants. The Red-Green-Blue and Depth (RGB-D), and multispectral images of each plant were collected synchronously by a Kinect camera and an hyperspectral imager (SOC710: 710 series by surface optics corporation) at four AOVs with an angle interval of 90° in the same imaging room. Subsequently, the multispectral reflectance was registered with a Fourier transform-based approach using depth coordinates. With a Kinect pose estimation and self-calibration method, the multiview point cloud coordinate system was unified for a rough multiview point cloud registration. Finally, the ICP algorithm was used for precise multiview point cloud registration to construct a multispectral 3D point cloud model of a tomato plant. According to the multispectral point cloud models of the plant canopy, the vegetation indexes, namely, the NDVI, GNDVI, RVI, difference vegetative index ratio (NDVIR), chlorophyll green index (CIG), red-edge CIG (RCIG), normalized green index (NG), normalized red index (NR), and green RVI (GRVI), were calculated and combined with the chlorophyll SPAD values to establish highly efficient prediction models and to measure the chlorophyll content in the plant canopy accurately. This measurement method can be expanded to assess the physiological features of other plant canopies. In summary, this method provides good technical support for high-throughput plant phenotypic analysis and is significant for the development of plant phenomics and other research fields.

## 2. Materials and Methods

### 2.1. Sample Cultivation

On 5 March 2019, 60 tomato seedlings (Fenguan No. 1) were transplanted into a Venlo greenhouse at the Modern Controlled-Environment Agricultural Technology and Equipment Engineering Laboratory in Jiangsu Province. Galuku coir substrate and Yamazaki tomato formula nutrient solution (pH 6.0–6.5, E_C_ 1.2 mS·cm^−1^) were used for cultivation. To manipulate the plant chlorophyll levels, 60 tomato plant samples were divided into five groups that received five different doses of nutrient solution, i.e., 25%, 75%, 100%, 150%, or 200% of the standard formula [33], with 100% being the dose required for normal tomato growth [34]; each group included 12 plants. From the 5th to 6th week after transplantation, the chlorophyll SPAD readings of the tomato leaves, as well as the plant multispectral and RGB-D images, were obtained.

### 2.2. Instrument and Chlorophyll Content Measurement

The multispectral 3D imaging system established in this study for greenhouse tomato plants consisted of an imaging room, a light source, an SOC710 hyperspectral imager, a Kinect sensor, an electric rotary table, a controller, and a graphics workstation (Figure 1a). The imaging room was made from aluminum profiles and its inner dimensions measured 180 (length) × 120 (width) × 160 cm (height); the interior sidewalls and floor were covered with white matte film. The power of the light source was 150 W, the response wavebands ranged from 380 to 2200 nm, and the intensity of the light source was stepless adjustable. The SOC710 hyperspectral imager had a built-in scanning system and covered a spectral range from 400 to 1000 nm for 128 wavebands; the image resolution was 696 × 510 px and the wavebands for imaging were selectable. The Kinect sensor (Microsoft Kinect 2.0) consisted of a color camera and a depth sensor. The RGB and depth images had resolutions of 1920 × 1080 px and 512 × 424 px, respectively; the frame rate was 30 fps, and the detection range was 0.50–4.50 m. The electric rotary table disc, which was 20.00 cm in diameter and was driven by a 57BYG stepper motor, could be rotated over a 360° range with a resolution of 0.0005° and a positioning precision of 0.01°. The controller used was an HW-36MT-3PG programmable controller with 20 input interfaces, 16 output interfaces, an RS232C communication interface, and a built-in 3-channel, 100 kHz, high-speed pulse output. The graphics workstation processor was an Intel (R) Xeon (R) E-2176M CPU @2.70 GHz with a 32G memory capacity and an NVIDIA Quadro P600 4G graphics card. Visual Studio 2015 and MATLAB 2017a were jointly used to program the system control.

Figure 1b–d shows a representative spectral image taken by the SOC710 hyperspectral imager (using 466.93 nm as an example) and RGB and depth images obtained by the Kinect sensor. The multiview images collected by the SOC710 hyperspectral imager and the Kinect sensor were used to reconstruct the multispectral 3D point cloud model of the tomato plants using a multispectral 3D reconstruction algorithm.

In this study, a portable chlorophyll meter (Konica Minolta SPAD-502 Plus; Minolta, Osaka, Japan) was used to measure SPAD values ranging from −9.9 to 199.9 SPAD, with a precision of ±1.0 SPAD and a repeatability of ±0.3 SPAD. The SPAD value was measured next to the vein on the widest part of the tomato leaf surface. Each tomato canopy was divided into upper, middle, and lower layers and the SPAD values were measured at ten sites for each layer; a total of 30 SPAD readings were obtained and averaged to derive the final SPAD value for each plant.

### 2.3. Multispectral 3D Point Cloud Modeling

Figure 2 presents a flowchart of the multispectral 3D imaging system for greenhouse tomato plants. Step 1 was used to initialize the system parameters and included the selection of the characteristic wavebands: Blue, Green, Red, Red-edge, and Near infrared (Nir), by the SOC710 hyperspectral imager and the determination of (i) the intrinsic parameters for the Kinect sensor, such as the principal point coordinates (*c*_x_, *c*_y_) and focal length (*f*_x_, *f*_y_), (ii) the area of the plant region of interest (ROI) (row, width, and height), and (iii) the number of AOVs (*V*_N_) required for plant 3D modeling. Step 2 was the Kinect pose estimation and self-calibration process. RGB-D images of the electric rotary table surface were taken at two or more AOVs and used to determine the point cloud coordinates of the yellow and red calibration stickers on the table surface at different AOVs according to the color threshold value. Subsequently, the center points of the two calibration stickers were identified to calculate the center coordinates and the normal vector of the rotation axis on the rotary table. Step 3 was spectral reflectance registration and calibration. The registration transformation matrix was calculated on the basis of the Fourier transform, then the spectral images obtained by the SOC710 imager were registered to the RGB image coordinate system obtained by the Kinect sensor. The spectral reflectance of multiple channels was mapped to the corresponding depth coordinates by using the same transformation parameters. Step 4 was the acquisition of the plant multiview RGB-D images and multispectral images. The plant RGB-D images and multispectral images were taken at all the desired AOVs by turning the electric rotary table for 360°/*V*_N_ after imaging at each AOV. Step 5 was the plant multispectral 3D reconstruction. According to the intrinsic parameters of the Kinect sensor, the RGB-D image at each AOV was converted into 3D point clouds. Subsequently, in accordance with the plant ROI, the 3D point clouds were pretreated by removing bounding boxes and outliers. Then, according to the center coordinates and normal vectors of the rotation axis on the rotary table, the coordinate system of the point cloud at each AOV was uniformly transformed, followed by the ICP registration of the point clouds at each AOV. Finally, the 3D point clouds were down-sampled to complete the plant 3D reconstruction, which also led to the successful reconstruction of the multispectral 3D point cloud model, because each spatial coordinate point contained information regarding the RGB value and the reflectance in each waveband.

#### 2.3.1. Spectral Reflectance Registration

When using both SOC710 images and Kinect images, one major task is to solve the transformation problems regarding the displacement, rotation angle, and zoom level. It is very difficult to obtain registration parameters by using an image feature point method (such as Scale-invariant feature transform [35], Speeded up robust feature [36], or Harris corner feature [37]) because (i) the image resolution of SOC710 is 696 × 510 px, the plant image area resolution of the Kinect sensor is 170 × 170 px, and the two imaging regions have large zoom multiples, (ii) the imaging regions of the two sensors are inconsistent, and (iii) the plant morphology is complex. In this study, the multispectral reflectance was registered to the depth coordinates using the principle of Fourier transform. In this approach, the translation, rotation matrix, and scaling coefficient between the two types of images are calculated according to the Fourier spectra of the images [38]. The frequency–domain relationship is shown in Formula (1), where the translation between the Kinect sensing image *F* (*x*, *y*) and the SOC710 multispectral image *M* (*x*, *y*) is (*x*_0_, *y*_0_), i.e., *F* (*x*, *y*) = *M* (*x* − *x*_0_, *y* − *y*_0_). As shown in Formula (2), the phase information on the cross-power spectrum includes the phase difference in the two images; through an inverse Fourier transformation of the cross-power spectrum, the 2D impact function *δ* (*u* − *x*_0_, *v* − *y*_0_) is obtained, which reaches its peak at (*x*_0_, *y*_0_) and is close to zero at the other positions. Finally, the translation parameter can be determined by identifying the peak.
(1)$$F(\xi ,\eta )=e^{-j2\pi (\xi x_{0}+\eta y_{0})}M(\xi ,\eta ),$$
(2)$$\frac{M(\xi ,\eta )F^{*}(\xi ,\eta )}{\left|M(\xi ,\eta )F^{*}(\xi ,\eta )\right|}=e^{-j2\pi (\xi x_{0}+\eta y_{0})},$$
where F(ξ,η) represents the Fourier transform of F(x,y), M(ξ,η) represents the Fourier transform of M(x,y), and $F^{*}(\xi ,\eta )$ represents the conjugate of F(ξ,η).

Resolving the transformation problems in rotation and scaling between the SOC710 and Kinect images requires the transformation of the rotation and scaling relationship in the Cartesian coordinates into a translation relationship in the log–polar coordinates using the Fourier–Mellin transformation. If it is assumed that *F* (*x*, *y*) is obtained by rotating image *M* (*x*, *y*) by *θ*_0_ and that the scaling coefficient is *σ*, then the rotation in the frequency domain of the image can be determined by using Formula (3) because of the invariance and similarity features in the Fourier rotation. Subsequently, Formula (4) is used to map the frequency–domain coordinates to the log–polar coordinate space. Formula (5) is used to convert the rotation and scaling into a translation relationship and the rotation angle *θ*_0_ and scaling coefficient *σ* can be calculated.
(3)$$\begin{array}{ll} F(x,y)= & M[\sigma (x\cos\theta_{0}+y\sin\theta_{0}),\\ & \sigma (-x\sin\theta_{0}+y\cos\theta_{0})] \end{array}$$
(4)$$\begin{array}{ll} F(\xi ,\eta )= & \frac{1}{\sigma^{2}}M[\sigma^{-1}(x\cos\theta_{0}+y\sin\theta_{0}),\\ & \sigma^{-1}(-x\sin\theta_{0}+y\cos\theta_{0})] \end{array}$$
(5)$$F(\log\rho ,\theta )=M(\log\rho -\log\sigma ,\theta -\theta_{0}),$$
where *σ* is the scaling coefficient, *θ*_0_ is the rotation angle, (*x*, *y*) are the image coordinates, F(logρ,θ) results from mapping F(ξ,η) in the log–polar coordinate space, $\rho^{2}=\xi^{2}+\eta^{2}$, and tanθ=ηξ.

Figure 3a shows the to-be-registered ROI *F* (*x*, *y*) as a grayscale Kinect image. Figure 3b presents a grayscale SOC710 image *M* (*x*, *y*) and Figure 3c presents the image after the registration of *M* (*x*, *y*). Figure 3d shows an ROI region spectral reflectance of 466.93 nm, which reveals that the remarkable difference only appears in a small portion of the marginal region. This finding suggests that the spectral reflectance registration was successful.

It is unnecessary to calibrate the transformation matrix repeatedly if the relative position relationship between the SOC710 hyperspectral imager and the Kinect sensor is fixed. In this study, the same transformation parameters were used for all spectral reflectance images in each waveband to maintain the consistency of spectral reflectance mapping. Following the spectral registration, the RGB image, depth image, and registered multiband reflectance image of each plant sample were saved in a 3D array to lay the foundation for reconstructing a 3D point cloud model of the plant.

#### 2.3.2. 3D Reconstruction of Multiview RGB-D Images

Figure 4a presents the electric rotary table used in this study, which had two circular calibration stickers (yellow and red, both 5 cm in diameter) on its round, blue surface. The Kinect pose estimation and self-calibration process was as follows. First, the RGB-D images of the rotary table were obtained and converted into 3D point clouds. According to the color threshold of the point clouds, the point cloud coordinates of the areas covered by the two calibration stickers were identified and the point cloud coordinates of the center of each sticker were calculated. Figure 4b demonstrates the coordinates of the two sticker centers, with Y1 representing the yellow sticker and R1 representing the red sticker, when the rotatory table was rotated by 0°. Similarly, Figure 4c shows the coordinates of the two sticker centers (Y2 and R2) when the rotatory table was rotated by 180°. Subsequently, according to the center coordinates of the calibration stickers in the two RGB-D images, the coordinates of the center *M* for the rotation axis of the rotary table were obtained. Finally, according to the coordinates of the calibration sticker centers in the two RGB-D images and the center coordinates *M* (*a*_0_, *b*_0_, *c*_0_) of the rotation axis, the normal vector *P* of the rotation axis of the rotary table was calculated and normalized to *P* (*a*, *b*, and *c*) (Figure 4d).

The rough registration stage: First, according to the intrinsic parameters of the Kinect sensor, i.e., the principal point coordinates (*c*_x_, *c*_y_) and focal length (*f*_x_, *f*_y_), the RGB-D images at various AOVs were converted into 3D point cloud images using Formula (6), in which the plant ROI area was identified from the determined ROI range. Subsequently, according to the center coordinates of the rotation axis *M* (*a*_0_, *b*_0_, *c*_0_), the translation transformation of the point cloud (*x_i_*, *y_i_*, *z_i_*) at each AOV was performed and the center of the rotation axis in the point cloud was shifted to the origin point (0, 0, 0) of the Kinect coordinate system, as shown in Formula (7). Next, according to the normal vector *P* (*a*, *b*, and *c*) of the rotation axis, the translated point cloud (*x_i_*′, *y_i_*′, *z_i_*′) was rotated until its normal vector reached the *Y*-axis, as shown in Formulas (8) and (9). Finally, using Formula (10), after the inverse rotation angle γ° around the *Y*-axis was determined according to the actual rotation angle of the image at each AOV, the coordinates of the point cloud (*X_i_*, *Y_i_*, *Z_i_*) for the unified coordinate system were obtained.

The precise registration stage: The ICP algorithm was used to obtain a precise multiview point cloud registration [39] in this study. First, ICP registration was performed for the point cloud (*X_i_*, *Y_i_*, *Z_i_*) at each AOV, that is, the point cloud at the first AOV was precisely ICP-registered with the point cloud at the adjacent second AOV. Then, the registration result was registered with the point cloud at the adjacent third AOV; the procedure was repeated until all the point clouds were registered to obtain a 3D point cloud model of the plant, namely, the 3DROI model.
(6)$$\left[\begin{array}{c} x_{i}\\ y_{i}\\ z_{i} \end{array}\right]=\left[\begin{array}{c} (i-c_{x})\times z_{i}/f_{x}\\ -(j-c_{y})\times z_{i}/f_{y}\\ Depth(i,j)/1000 \end{array}\right],$$
(7)$$\left[\begin{array}{c} x_{i}^{\prime }\\ y_{i}^{\prime }\\ z_{i}^{\prime }\\ 1 \end{array}\right]=\left[\begin{array}{cccc} 1 & 0 & 0 & -a_{0}\\ 0 & 1 & 0 & -b_{0}\\ 0 & 0 & 1 & -c_{0}\\ 0 & 0 & 0 & 1 \end{array}\right]\left[\begin{array}{c} x_{i}\\ y_{i}\\ z_{i}\\ 1 \end{array}\right],$$
(8)$$\left[\begin{array}{c} x_{i}^{\prime \prime }\\ y_{i}^{\prime \prime }\\ z_{i}^{\prime \prime }\\ 1 \end{array}\right]=\left[\begin{array}{cccc} 1 & 0 & 0 & 0\\ 0 & \cos\alpha & -\sin\alpha & 0\\ 0 & \sin\alpha & \cos\alpha & 0\\ 0 & 0 & 0 & 1 \end{array}\right]\left[\begin{array}{c} x_{i}^{\prime }\\ y_{i}^{\prime }\\ z_{i}^{\prime }\\ 1 \end{array}\right]=\left[\begin{array}{cccc} 1 & 0 & 0 & 0\\ 0 & b/\sqrt{b^{2}+c^{2}} & -c/\sqrt{b^{2}+c^{2}} & 0\\ 0 & c/\sqrt{b^{2}+c^{2}} & b/\sqrt{b^{2}+c^{2}} & 0\\ 0 & 0 & 0 & 1 \end{array}\right]\left[\begin{array}{c} x_{i}^{\prime }\\ y_{i}^{\prime }\\ z_{i}^{\prime }\\ 1 \end{array}\right],$$
(9)$$\left[\begin{array}{c} x_{i}^{\prime \prime \prime }\\ y_{i}^{\prime \prime \prime }\\ z_{i}^{\prime \prime \prime }\\ 1 \end{array}\right]=\left[\begin{array}{cccc} \cos\beta & \sin\beta & 0 & 0\\ -\sin\beta & \cos\beta & 0 & 0\\ 0 & 0 & 1 & 0\\ 0 & 0 & 0 & 1 \end{array}\right]\left[\begin{array}{c} x_{i}^{\prime \prime }\\ y_{i}^{\prime \prime }\\ z_{i}^{\prime \prime }\\ 1 \end{array}\right]=\left[\begin{array}{cccc} \sqrt{b^{2}+c^{2}}/\sqrt{a^{2}+b^{2}+c^{2}} & a/\sqrt{a^{2}+b^{2}+c^{2}} & 0 & 0\\ -a/\sqrt{a^{2}+b^{2}+c^{2}} & \sqrt{b^{2}+c^{2}}/\sqrt{a^{2}+b^{2}+c^{2}} & 0 & 0\\ 0 & 0 & 1 & 0\\ 0 & 0 & 0 & 1 \end{array}\right]\left[\begin{array}{c} x_{i}^{\prime \prime }\\ y_{i}^{\prime \prime }\\ z_{i}^{\prime \prime }\\ 1 \end{array}\right],$$
(10)$$\left[\begin{array}{c} X_{i}\\ Y_{i}\\ Z_{i}\\ 1 \end{array}\right]=\left[\begin{array}{cccc} \cos\gamma & 0 & -\sin\gamma & 0\\ 0 & 1 & 0 & 0\\ \sin\gamma & 0 & \cos\gamma & 0\\ 0 & 0 & 0 & 1 \end{array}\right]\left[\begin{array}{c} x_{i}^{\prime \prime \prime }\\ y_{i}^{\prime \prime \prime }\\ z_{i}^{\prime \prime \prime }\\ 1 \end{array}\right],$$
where (*i*, *j*) are the coordinates of the depth image, (*c*_x_, *c*_y_) are the principal point coordinates of the Kinect sensor, (*f*_x_, *f*_y_) are the focal lengths of the Kinect sensor, *Depth* (*i*, *j*) is the depth value at coordinates (*i*, *j*) measured in mm, (*x_i_*, *y_i_*, *z_i_*) are the coordinates of the point cloud at the *i*th AOV, (*x_i_*′, *y_i_*′, *z_i_*′) are the post-translation coordinates of the point cloud at the *i*th AOV, (*x_i_*‴, *y_i_*‴, *z_i_*‴) are the coordinates of the point cloud at the *i*th AOV after one rotation around the *X*-axis, (*x_i_*‴, *y_i_*‴, *z_i_*‴) are the coordinates of the point cloud at the *i*th AOV after one rotation around the *Z*-axis, (*X_i_*, *Y_i_*, *Z_i_*) are the coordinates of the point cloud at the *i*th AOV in the unified coordinate system, *α* is the angle between the *Y*-axis and the normal vector projection on the YOZ plane, measured in °, *β* is the angle between the *Y*-axis and the normal vector projection on the XOY plane, measured in °, *γ* is the rotation angle of the point cloud relative to the reference AOV (i.e., the first AOV for image acquisition in this study), measured in °, (*a*_0_, *b*_0_, *c*_0_) are the center coordinates of the rotation axis, and (*a*, *b*, *c*) is the normalized normal vector of the rotation axis on the rotary table.

### 2.4. Data Processing

This study included 60 tomato plant samples. For each plant, four RGB-D images were taken with a Kinect sensor at four AOVs with an interval angle of 90°. The four AOVs are represented by AOV1, AOV2, AOV3, and AOV4. The multispectral reflectances at the four AOVs were synchronously collected by an SOC710 imager and a Kinect sensor at the bandwidths of 466.93 (ρ_Blue_), 564.91 (ρ_Green_), 696.32 (ρ_Red_), 722.94 (ρ_Red__-edge_), and 841.4 nm (ρ_Nir_), which are the characteristic bands to which the nitrogen content in tomato plants were sensitive and therefore used as an example [6,7,30]. By following the method described in Section 2.3, we reconstructed the multispectral point cloud 3D models of the 60 tomato plant samples and we calculated the mean, standard deviation, number of points, and coefficient of variation of the spectral reflectance for the plant canopy in the AOV1, AOV2, AOV3, AOV4, and 3DROI point cloud models. In addition, a series of vegetation indexes was calculated from the plant canopy multispectral 3D point cloud ROI model; these formulae are shown in Table 1. Prediction models that used various vegetation indexes and chlorophyll SPAD values were established and their coefficients of determination (*R*^2^) and root-mean-square-errors (*RMSE*) were calculated. The prototype functions of the prediction models included the linear function, quadratic function, exponential function, and power function, which were expressed as M1, M2, M3, and M4, respectively.

## 3. Results and Analysis

### 3.1. Multispectral 3D Point Cloud Modeling

From the determined ROI range, the tomato plant area was identified and the background area was considered “not a number” (NaN). Figure 5a–d shows the depth images at AOV1, AOV2, AOV3, and AOV4, respectively. The spectral reflectance registration procedures described in Section 2.3.1 were used to register all the multispectral reflectance images at different AOVs in depth image coordinate systems. Figure 5e–h shows the spectral reflectance for the AOV1, AOV2, AOV3, and AOV4 images after the registration, respectively (with *ρ*_Blue_ as an example). The RGB-D image 3D reconstruction method described in Section 2.3.2 was used to convert the depth images at all the AOVs into point cloud images. Their multispectral reflectance values were *ρ*_Blue_, *ρ**_G_*_reen_, *ρ*_Red_, *ρ*_Red__-edge_, and *ρ*_Nir_. Figure 5i–l presents the 3D point clouds for the AOV1, AOV2, AOV3, and AOV4 images with the unified coordinate system (with *ρ*_Blue_ as an example). As a result of the morphological complexity of the plant canopy and the shielding effect of the leaves, it is impossible to establish a complete plant point cloud model from a single AOV; hence, it is necessary to integrate multiview point cloud models. In this study, the 3D point cloud models of the tomato plants at the four AOVs were obtained and used to reconstruct a multispectral 3D point cloud image through precise ICP registration. Figure 5m–q presents the 3D point cloud models of *ρ*_Blue_, *ρ*_Green_, *ρ*_Red_, *ρ*_Red__-edge_, and *ρ*_Nir_.

In this study, the SOC710 hyperspectral imager captured the image of the plant parts above the pot and the Kinect sensor captured the image of the entire plant area, including the pot and the background area. Because the range differs between the two types of images, the multiband spectral reflectance information was available only for the plant canopy area and not the plant part below the top edge of the pot. Therefore, the region with the missing spectral reflectance value was considered “not a number” (NaN). The ROI range was determined on the basis of the imaging distance and the relative positions of the sensor and lens parameters, which were appropriately selected according to the actual requirement for multispectral 3D reconstruction.

### 3.2. Spectral Reflectance Variability Analysis

In this study, the center coordinates of the rotation axis for the electric rotary table were (0, 0, 0) and the 3D point clouds of the plant canopy area were identified according to the height of the pot. The mean value, standard deviation (STD), and coefficient of variation (CV) of the spectral reflectance for the AOV1, AOV2, AOV3, AOV4, and canopy 3DROI point cloud models of 60 tomato plants in the Blue, Green, Red, Red-edge, and Nir characteristic wavebands were calculated (Figure 6a,b). In addition, the numbers of points in the AOV1, AOV2, AOV3, AOV4, and 3DROI canopy models were counted (Figure 6c).

As shown in Figure 6a, because the 3D geometric morphological of the tomato plant canopy is complex, the plant ROI image varied at different AOVs, leading to significant variations in the spectral reflectance (Figure 5e–h). In this study, the STD was used to evaluate the degree of deviation in the spectral reflectance within the ROI region of the plants and the CV was used to assess the mean spectral reflectance variability among the AOV1, AOV2, AOV3, AOV4, and 3DROI point cloud models.

The maximum value (MAX), minimum value (MIN) and average value (AVG) of spectral reflectance STD of the AOV1, AOV2, AOV3, AOV4, and 3DROI point cloud models was calculated for all 60 tomato plants in the Blue, Green, Red, Red-edge, and Nir characteristic wavebands. As shown in Table 2, for the Blue, Green, Red, Red-edge, and Nir wavebands, the mean STD values of the spectral reflectance in the AOV1, AOV2, AOV3, AOV4, and 3DROI point cloud models of the plant canopy had ranges of 0.1138–0.1193, 0.1046–0.1088, 0.1143–0.1200, 0.0944–0.1129, and 0.0870–0.1184, with maximum standard deviations of 0.1823, 0.1814, 0.1772, 0.2107, and 0.2889, respectively. This finding indicates that, regardless of whether the single-view or multiview reconstruction point cloud models was used, the spectral reflectance distribution of the plant canopy varied among the different spatial coordinates. Therefore, it is impossible to use multispectral reflectance under a single spatial coordinate system to construct an accurate SPAD prediction model. In addition, the variations in the chlorophyll distribution of the plant canopy should be considered.

In the Blue, Green, Red, Red-edge, and Nir wavebands, the AOV1, AOV2, AOV3, AOV4, and 3DROI point cloud models of the 60 tomato plants had minimum spectral reflectance CVs of 3.25%, 0.97%, 2.09%, 0.67%, and 0.43%, maximum CVs of 16.95%, 12.03%, 22.08%, 23.00%, and 14.50%, and average CVs of 7.48%, 4.50%, 6.75%, 4.94%, and 3.73%, respectively (Figure 6b). This finding indicates that there were certain variations in the spectral reflectance among the AOV1, AOV2, AOV3, AOV4, and 3DROI point cloud models and the differences were remarkable for different AOVs in some plants. Therefore, a SPAD value prediction model constructed using spectral reflectance data from single-view point cloud models would yield unstable results because of the variations among different AOVs.

Figure 6c shows the number of points in the point cloud models of the 60 tomato plants for AOV1, AOV2, AOV3, AOV4, and 3DROI. The mean numbers were 5695.83, 5226.22, 5063.57, 5421.18, and 13,782.83 for the AOV1, AOV2, AOV3, AOV4, and 3DROI models, respectively. Although the number of points in each single-view point cloud model did not vary significantly, there were far fewer points compared with the number of points in the multiview point cloud model reconstructed from the four AOV models. On average, the number of points in the AOV1, AOV2, AOV3, and AOV4 point cloud models respectively accounted for 43.23%, 39.39%, 38.08%, and 41.06% of the number of points in the 3DROI point cloud model. That is, overall, the number of points for all the single-view point cloud models accounted for a minimum of 21.62%, a maximum of 70.79%, and an average of 40.47% of the number in the multiview reconstructed point cloud model. Therefore, the spectral reflectance information for the plant canopy in a single-view model cannot fully reflect the actual status of the canopy and it is neither scientific nor rigorous enough to construct a plant canopy chlorophyll prediction model with only single-view spectral reflectance data.

In this study, the statistical analysis of characteristic variables in the plant canopy chlorophyll measurement model revealed that there were significant differences in the spectral reflectance distribution in the single-view point cloud models, as well as in the spectral reflectance value and the number of points among different AOV point cloud models. In addition, the number of points in the single-view point cloud models accounted for an average of only 40.47% of that in the 3D reconstructed point cloud model. This finding suggests that a single-view point cloud model cannot fully reflect the status of the characteristics of the spectral reflectance in a plant canopy. Therefore, this study proposes the construction of a prediction model for plant chlorophyll SPAD with the characteristic variables of multiview, multispectral point cloud models.

### 3.3. Plant Chlorophyll Measurement Model and Analysis

In this study, the average value of the vegetation index obtained from the 3D multispectral point cloud model of the plant canopy was employed as the characteristic input variable of the measurement model. The vegetation indexes used for the analysis were the NDVI, GNDVI, NDVIR, CIG, RCIG, NG, NR, RVI, and GRVI (Table 1). The measured mean value of plant chlorophyll was taken as the output variable of the model. The prototype functions used in the measurement model were the linear function, quadratic function, exponential function, and power function, which were represented by M1, M2, M3, and M4, respectively. Prediction models that each used a different vegetation index and the chlorophyll SPAD value were established and their R^2^ and RMSE values were calculated. Figure 7a–i presents the 3D multispectral point cloud models established by using NDVI, GNDVI, NDVIR, CIG, RCIG, NG, NR, RVI, and GRVI, respectively.

Figure 8a,b shows the *R*^2^ and RMSE values of the 180 regression models that were established using the SPAD value and nine vegetation indexes (NDVI, GNDVI, NDVIR, CIG, RCIG, NG, NR, RVI, and GRVI) from the AOV1, AOV2, AOV3, AOV4, and 3DROI point cloud models. The *R*^2^ and the RMSE values ranged from 0.7279 to 0.9443 and from 0.8505 to 1.8471, respectively.

The comparison of the determination coefficient *R*^2^ values among the regression models with different prototype functions (Figure 8c) reveals that the mean *R*^2^ value of the regression model with the quadratic function (M2) as the prototype function was larger than that of the models using the linear function (M1), the exponential function (M3), or the power function (M4) as the prototype function. However, the difference in the overall regression/correlation was insignificant among all the M1–M4 regression models. This insignificance might be attributable to the small range of plant SPAD readings. In this study, the SPAD readings ranged from 31.46 to 60.90 and the average value ranged from 37.49 to 53.35. When the vegetation indexes from the 3DROI point cloud model were used as the input variables to construct the M1, M2, M3, and M4 regression models, the mean *R*^2^ values were 0.8995, 0.9041, 0.9008, and 0.8968 and the mean *RMSE* values were 1.1016, 1.0733, 1.0933, and 1.1206, respectively. The *R*^2^ of the regression model using the vegetation indexes from the 3DROI point cloud model was larger than that of the regression model using the vegetation indexes from the single-view point cloud models. This finding indicated that the chlorophyll measurement model that was established by using characteristic variables from the 3DROI point cloud model had a higher correlation regardless of the prototype function that was used for the regression analysis.

Figure 8d compares the mean determination coefficient *R*^2^ values of all four regression methods among the models with different vegetation indexes as the input variable. When the NDVI, GNDVI, NDVIR, CIG, RCIG, NG, NR, RVI, or GRVI from the 3DROI point cloud model were used as the input variables, the mean *R*^2^ values were 0.8641, 0.9390, 0.9222, 0.9418, 0.9002, 0.9022, 0.8027, 0.8945, and 0.9362 and the mean RMSE values were 1.3085, 0.8826, 1.003, 0.8623, 1.1133, 1.0950, 1.5927, 1.1409, and 0.8903, respectively. When the vegetation index from the 3DROI point cloud model was used for the regression, the determination coefficient *R*^2^ was larger than that obtained when the vegetation index from the single-view point cloud model was used. Furthermore, the highest *R*^2^ resulted from the regression model using CIG as the input, followed by GNDVI, GRVI, NDVIR, NG, RCIG, RVI, NDVI, and NR. Table 3 presents the SPAD regression equations and the results of using the NDVI, GNDVI, NDVIR, CIG, RCIG, NG, NR, RVI, or GRVI from the 3DROI point cloud model as the input variables.

In summary, the vegetation indexes obtained from the 3DROI point cloud model can be used to predict the chlorophyll SPAD value and evaluate the spatial SPAD distribution in the plant canopy. The performance of the regression model that was constructed from 3DROI data is superior to those using single-view data. Because its accuracy and stability are high, the proposed method is a promising tool that can be expanded for canopy physiological assessments of other plants.

## 4. Conclusions

A multispectral, 3D, nondestructive measurement system for greenhouse tomato plants was designed and a heterogeneous image registration method based on the Fourier transform was proposed. The values of the translation, rotation angle, and scaling coefficient between the SOC710 hyperspectral and Kinect images were used to accurately register the multispectral reflectance information to the depth coordinate system, which laid the foundation for reconstructing a multispectral 3D point cloud model. A 3D reconstruction method using multiview RGB-D images was developed using Kinect pose estimation and self-calibration to obtain a unified transformation matrix for multiview RGB-D image coordinate systems, which solved the fast rough registration of multiview point clouds. Finally, a multispectral 3D point cloud model of the plant was reconstructed using a precise ICP registration algorithm. The vegetation index from the multispectral 3D point cloud model was used as the input variable and the measured SPAD value was the output, resulting in an SPAD measurement model of the plant canopy chlorophyll. The performance of the SPAD measurement model, based on the 3D multispectral point cloud model and single-view point cloud model, was compared and analyzed by a plant canopy chlorophyll measurement experiment. The results revealed that the measurement model that was established by using the characteristic variables from a multiview point cloud model was superior to the one established using the variables from a single-view point cloud model and obtained a precise and efficient high-throughput measurement of plant chlorophyll.

In the present study, only a single-frame RGB-D image was used to establish 3D point cloud models at each AOV; therefore, the accuracy of the point cloud modeling was limited by the Kinect sensor. This drawback could be overcome by using multiframe RGB-D images to construct point cloud models at each AOV. The integration of multiple point cloud models at each AOV might improve the performance of point cloud reconstruction for small plant stems, thereby increasing the precision of 3D point cloud modeling. As a result, the model performance for plant SPAD prediction can be further improved and can provide more precise information for optimizing the plant canopy SPAD spatial distribution and density. The multispectral 3D reconstruction approach for the greenhouse tomato plants used in this study can be used to reconstruct the plant multispectral 3D point cloud model, which optimizes the traditional 2D image measurement method and can serve as a precise and efficient tool for high-throughput plant chlorophyll measurements. This low-cost, easy-to-operate method can be expanded to the physiological assessments of other plant canopies. This approach is promising and of great significance to the development of plant phenomics and other research fields.

## Figures and Tables

**Figure 1 sensors-19-03345-f001:**
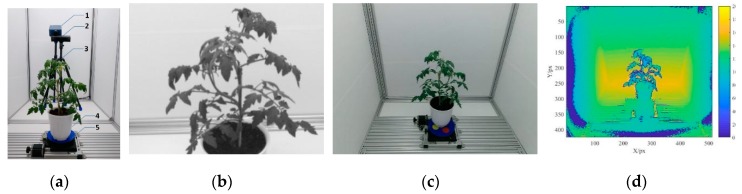
Multispectral three-dimensional reconstruction system for the greenhouse tomato plants. (**a**) Multispectral three-dimensional reconstruction system (1. SOC710 hyperspectral imager; 2. Kinect senor; 3. Tripod; 4. Tomato plant; 5. Electric turntable). (**b**) Spectral image at 466.93 nm by the SOC710 senor. (**c**) RGB image by the Kinect sensor. (**d**) Depth image by the Kinect sensor.

**Figure 2 sensors-19-03345-f002:**
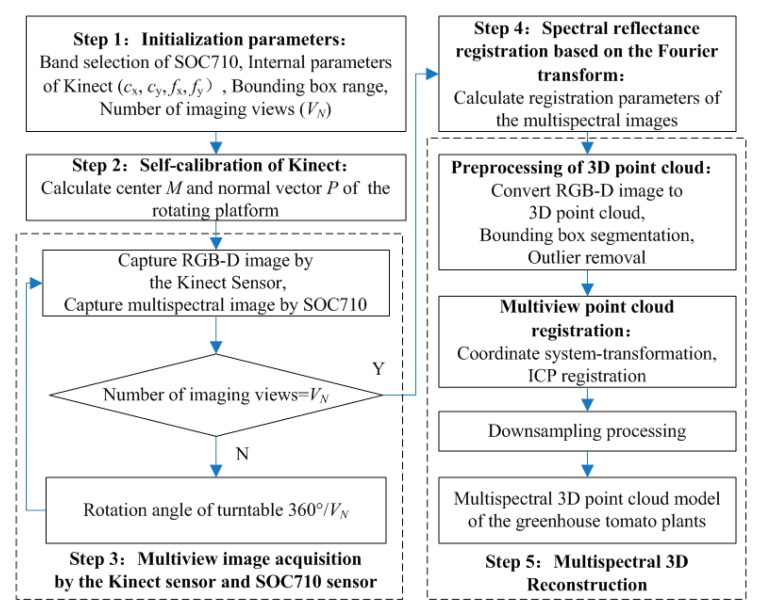
A flowchart for soil and plant analyzer development (SPAD) measurement of the greenhouse tomato plant canopy.

**Figure 3 sensors-19-03345-f003:**
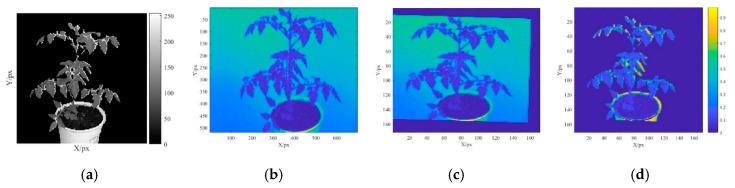
Spectral reflectance registration images. (**a**) Grayscale image of the calibration board (by SOC710), (**b**) grayscale image of the region of interest (ROI) on the calibration board (by Kinect), (**c**) image after SOC710 image registration, and (**d**) difference in the ROI.

**Figure 4 sensors-19-03345-f004:**
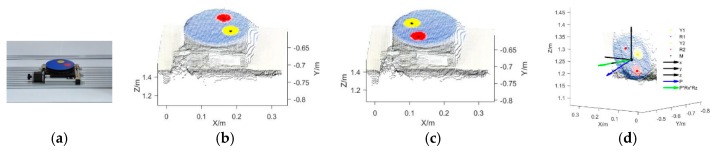
Pose estimation and self-calibration of the Kinect sensor. (**a**) Electric rotary table, (**b**) identification of the centers of the calibration stickers (rotation angle 0°), (**c**) identification of the centers of the calibration stickers (rotation angle 180°), and (**d**) center of the rotation axis and the normal vector.

**Figure 5 sensors-19-03345-f005:**
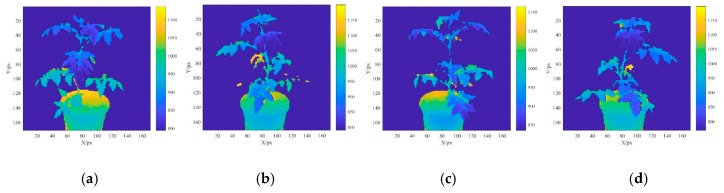
Reconstruction of multispectral 3D point cloud models of tomato plants. (**a**) ROI depth image of ROI at angle of view (AOV)1, (**b**) ROI depth image of ROI at AOV2, (**c**) ROI depth image of ROI at AOV3, (**d**) ROI depth image of ROI at AOV4, (**e**) spectral reflectance profile of ROI at AOV1, (**f**) spectral reflectance profile of ROI at AOV2, (**g**) spectral reflectance profile of ROI at AOV3, (**h**) spectral reflectance profile of ROI at AOV4, (**i**) point cloud image of ROI at AOV1, (**j**) point cloud image of ROI at AOV2, (**k**) point cloud image of ROI at AOV3, (**l**) point cloud image of ROI at AOV4, (**m**) point cloud model at 466.93 nm, (**n**) point cloud model at 564.91 nm, (**o**) point cloud model at 696.32 nm, (**p**) point cloud model at 722.94 nm, and (**q**) point cloud model at 841.42 nm.

**Figure 6 sensors-19-03345-f006:**
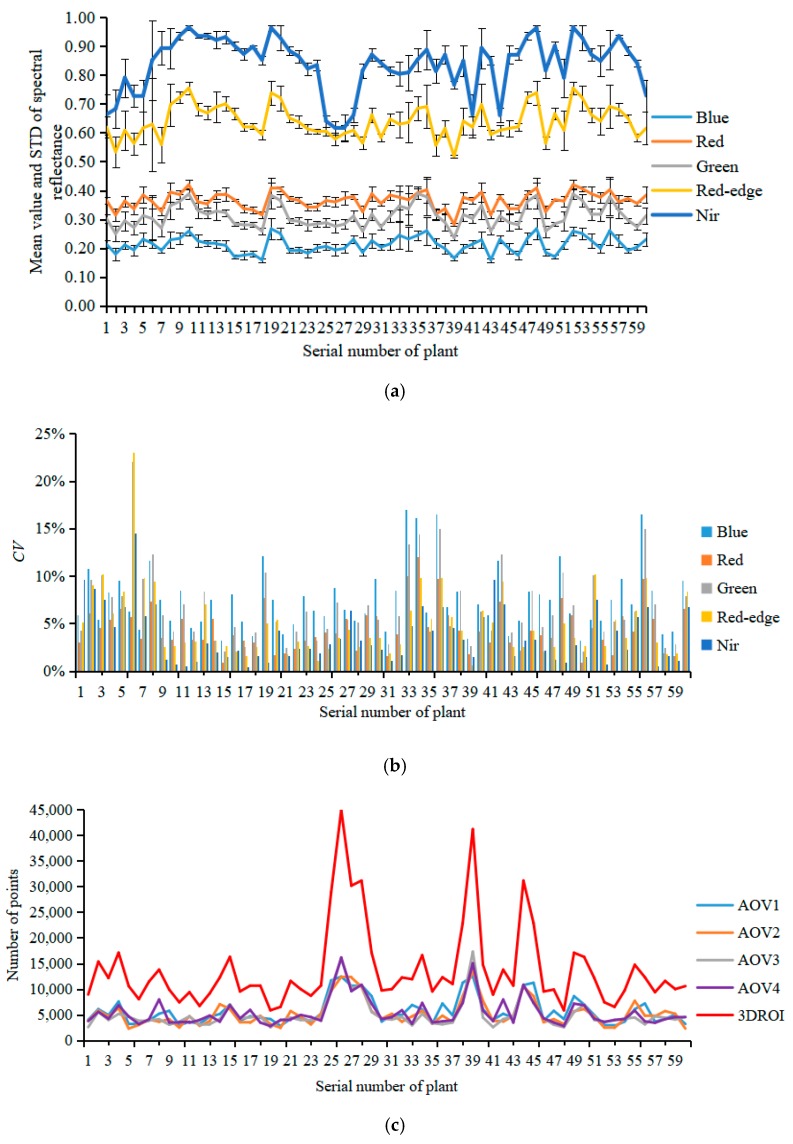
Analysis of spectral reflectance variability in multispectral point cloud models for the plant canopy. (**a**) Mean value and standard deviation of the spectral reflectance, (**b**) coefficients of variation of the spectral reflectance at different AOVs, (**c**) number of points in the point cloud models at different AOVs.

**Figure 7 sensors-19-03345-f007:**
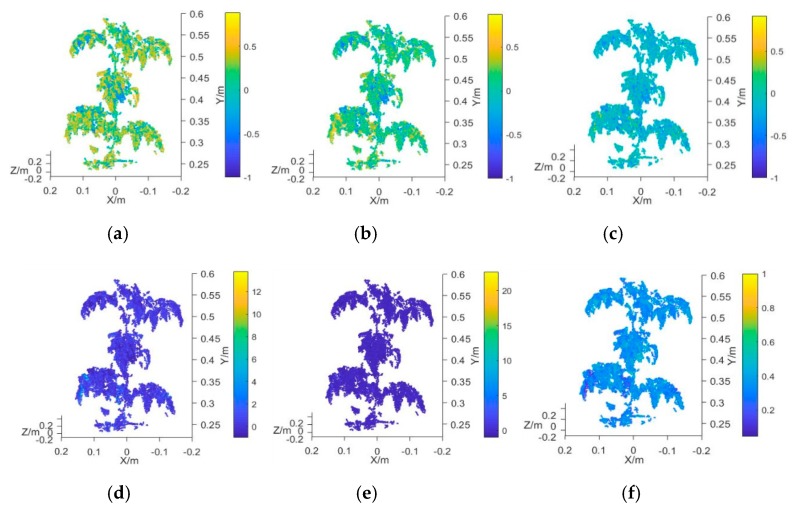
Plant canopy point cloud model using different vegetation indexes. (**a**) Point cloud model using NDVI, (**b**) point cloud mode using GNDVI, (**c**) point cloud mode using NDVIR, (**d**) point cloud mode using CIG, (**e**) point cloud mode using RCIG, (**f**) point cloud mode using NG, (**g**) point cloud mode using NR, (**h**) point cloud mode using RVI, and (**i**) point cloud mode using GRVI.

**Figure 8 sensors-19-03345-f008:**
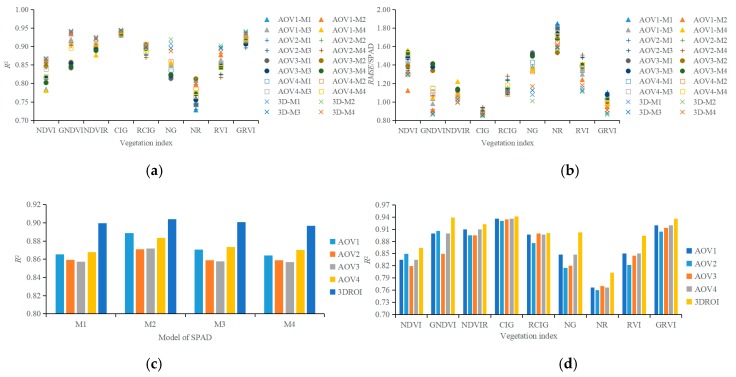
Performance of regression models used in plant canopy chlorophyll predictions. (**a**) *R*^2^ of regression models, (**b**) RMSE of regression models, (**c**) *R*^2^ when using different prototype functions to construct the regression models, and (**d**) *R*^2^ when using different vegetation indexes as input variable.

**Table 1 sensors-19-03345-t001:** Vegetation indexes.

Vegetation Index	Calculation Formula	Vegetation Index	Calculation Formula	Vegetation Index	Calculation Formula
*NDVI*	*NDVI* = (*ρ*_N_ir__ − *ρ*_R_ed__)/(*ρ*_N_ir__ + *ρ*_R_ed__)	*GNDVI*	*GNDVI* = (*ρ*_N_ir__ − *ρ*_G_reen__)/(*ρ*_N_ir__ + *ρ*_G_reen__)	*NDVIR*	*NDVIR* = (*ρ*_N_ir__ − *ρ*_R_ed____-____edge__)/(*ρ*_N_ir__ + *ρ*_R_ed____-____edge__)
*CIG*	*CIG* = *ρ*_N_ir__/*ρ*_G_reen__ − 1	*RCIG*	*RCIG* = *ρ*_N_ir__/*ρ*_R_ed____-____edge__ − 1	*NG*	*NG* = *ρ*_G_reen__/(*ρ*_N_ir__ + *ρ*_G_reen__ + *ρ*_R_ed__)
*NR*	*NR* = *ρ*_R_ed__/(*ρ*_N_ir__ + *ρ*_G_reen__ + *ρ*_R_ed__)	*RVI*	*RVI* = *ρ*_N_ir__/*ρ*_R_ed__	*GRVI*	*GRVI* = *ρ*_N_ir__/*ρ*_G_reen__

**Table 2 sensors-19-03345-t002:** Spectral reflectance standard deviation (STD) values.

Point Cloud Model	Blue			Green			Red			Red-edge			Nir		
	MIN	MAX	AVG	MIN	MAX	AVG	MIN	MAX	AVG	MIN	MAX	AVG	MIN	MAX	AVG
AOV1	0.0826	0.1453	0.1177	0.0757	0.1481	0.1078	0.0847	0.1509	0.1200	0.0489	0.2107	0.1102	0.0006	0.2718	0.1083
AOV2	0.0809	0.1539	0.1193	0.0741	0.1448	0.1057	0.0766	0.1603	0.1177	0.0403	0.1998	0.0944	0.0194	0.2794	0.0870
AOV3	0.0829	0.1823	0.1179	0.0768	0.1814	0.1056	0.0869	0.1772	0.1168	0.0592	0.2289	0.0996	0.0174	0.2680	0.0916
AOV4	0.0850	0.1787	0.1186	0.0785	0.1669	0.1088	0.0738	0.1633	0.1191	0.0515	0.2260	0.1047	0.0185	0.2889	0.1032
3DROI	0.0773	0.1598	0.1138	0.0703	0.1589	0.1046	0.0832	0.1591	0.1143	0.0584	0.2060	0.1129	0.0320	0.2569	0.1184

**Table 3 sensors-19-03345-t003:** SPAD Regression Equation and analysis results.

Vegetable Index	Point Cloud Model	Prototype Function	SPAD Regression Equation	*R* ^2^	*RMSE*
*NDVI*	*3DROI*	M2	SPAD = 154.8*NDVI*^2^ − 77.053*NDVI* + 43.256	0.8670	1.2916
*GNDVI*	*3DROI*	M3	SPAD = 14.998 × 10^2.4631*GNDVI*^	0.9414	0.8725
*NDVIR*	*3DROI*	M4	SPAD = 120.28*NDVIR*^0.5542^	0.9253	0.9940
*CIG*	*3DROI*	M3	SPAD = 24.001 × 10^0.333*CIG*^	0.9443	0.8508
*RCIG*	*3DROI*	M2	SPAD = −18.37*RCIG*^2^ + 58.963*RCIG* + 23.433	0.9023	1.1066
*NG*	*3DROI*	M2	SPAD = −2311.2*NG*^2^ + 762.19*NG* − 9.1868	0.9188	1.008
*NR*	*3DROI*	M3	SPAD = 122.99 × 10^−5.537*NR*^	0.8071	1.5918
*RVI*	*3DROI*	M2	SPAD = 1.3041*RVI*^2^ − 2.805*RVI* + 35.654	0.9019	1.1091
*GRVI*	*3DROI*	M2	SPAD = 5.9953*GRVI*^2^ − 19.913*GRVI* + 51.704	0.9408	0.8616

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
