# Peer review of "Measurement Method Based on Multispectral Three-Dimensional Imaging for the Chlorophyll Contents of Greenhouse Tomato Plants"

_sensors, 2019, doi:10.3390/s19153345_

Round 1

Reviewer 1 Report

The authors describe a method where they fuse the depth information data of a Kinect 2 and a multispectral camera to 3D point clouds to analyze the chlorophyll content of tomato plants. The topic is interesting and relevant for the field. However, there are some issues which should be addressed by the authors:

Abstract: The abstract does not provide a real structure, it is too long, has unnecessary information like all R² results e.g. The important information for understanding the paper are not present. Also it is not clear what is the improvement with 3D analysis compared to “traditional” 2D analysis.

Introduction:

There are missing descriptions of some state of the art, ICP use and reference are not mentioned. Other work using Kinect and ICP for plant reconstruction are not mentioned.

One example:

https://www.sciencedirect.com/science/article/pii/S0168169917309730

What is SPAD? Please explain

Line 100: there is Laser (LiDAR) or radar. This is not the same sensor. No reference for radar

Multi view stereo reconstruction is one option and the other refer to “photogrammetry” [25,26]

Line 109: SPAD was not explained jet

Line 115: what do you mean with “the same imaging room” ? unclear

Line 151: do you mean measurement range? Or do you mean the distance to the plant?

Figure 1: description of a) is doesn’t fit to the image

I don’t get why we need the Furrier Transform. Can you try to clarify this point?

How are the stickers detected? Manual selection? Algorithm? Color?

2.4. Data processing:

How did the ICP perform with 90° movement interval? I would suggest it failed in most cases to align as the ICP need at least 80% overlapping points to work. Did you use fixed angles instead? Did you perform ICP matching with more angles? Could you provide a 3D image of the resulting matched point cloud? Did you now just used 4 AOVs per plant or more views for the reconstruction?

Line 308: you don’t have to name every time all indexes you use. Please change this in the paper.

Figure 6 is useless please delete, or give result as pillow diagram of all plants as a mean value with error bars

How did you crop out the pot from the point cloud?

Figure 7: Z axis looks weird. 3D view possible? Could you show the registration result? Or is this just one shot?

How was the outcome of the different nutrition to the chlorophyll content of the tomato plants? Which tomatoes had which real value (ground truth)?

Conclusion is too long and mixed with discussion and results. Please shorten it to the necessary outcomes and the “golden nugget” of the paper

Author Response

Response to Reviewer 1 Comments

Dear Reviewers and Editor,

We have revised the manuscript based on the Reviewers’ comments, and each change made in response to these comments is marked in red font in the revised manuscript (attached). Below, please find our responses to Reviewers’ comments:

Comments and Suggestions for Authors

The authors describe a method where they fuse the depth information data of a Kinect 2 and a multispectral camera to 3D point clouds to analyze the chlorophyll content of tomato plants. The topic is interesting and relevant for the field. However, there are some issues which should be addressed by the authors:

Abstract: The abstract does not provide a real structure, it is too long, has unnecessary information like all R² results e.g. The important information for understanding the paper are not present. Also it is not clear what is the improvement with 3D analysis compared to “traditional” 2D analysis.

Response: 

The abstract has been revised and marked in red font.

Introduction:

There are missing descriptions of some state of the art, ICP use and reference are not mentioned. Other work using Kinect and ICP for plant reconstruction are not mentioned.

One example:

https://www.sciencedirect.com/science/article/pii/S0168169917309730

Response: 

The plant three-dimensional reconstruction methods based on Kinect sensor and ICP algorithm have been supplemented in the introduction. It is marked in red font and supplemented with relevant references.

Kinect sensor-based 3D plant reconstruction can be divided into single-view [28], and multiview reconstruction [29], the latter of which mainly uses the iterative closest point (ICP) algorithm [30, 31]. However, first of all, we need to solve the problem of rough registration of multiview point clouds, otherwise, ICP can not be used to accurate registration.

28. George, A.; Michael, L.; Radu, B. Rapid characterization of vegetation structure with a microsoft kinect sensor. Sensors 2013, 13, 2384-2398; DOI:10.3390/s130202384.

29. Dionisio, A.; César, F.; José, D. Matching the best viewing angle in depth cameras for biomass estimation based on poplar seedling geometry. Sensors 2015, 15, 12999-13011; DOI:10.3390/s150612999.

30. Manuel, V.; David, R.; Dimitris, S.; Miguel, G.; Marlowe, E. Hans, W. 3-D reconstruction of maize plants using a time-of-flight camera. Comput. Electron. Agric. 2018, 145, 235247.

31. Yang, H.; Le, W.; Lirong, X.; Qian, W.; Huanyu, J. Automatic non-destructive growth measurement of leafy vegetables based on kinect. Sensors 2018, 18, 806; DOI:10.3390/s18030806.

What is SPAD? Please explain

Response: 

SPAD is the abbreviation for Soil and Plant Analyzer Development, SPAD is a method of measuring the concentration of chlorophyll, which was invented by Japanese. SPAD is the method of measuring the concentration of chlorophyll by SPAD.

Line 100: there is Laser (LiDAR) or radar. This is not the same sensor. No reference for radar

Multi view stereo reconstruction is one option and the other refer to “photogrammetry” [25,26]

Response: 

Yes, the description in the manuscript is incorrect.

“3D laser radar” has been modified to “3D LiDAR” .   

“Multiview stereo reconstruction” has been modified to “multiview photogrammetry”.

Line 109: SPAD was not explained jet

Response: 

We added the explanation of SPAD in the Introduction.

SPAD is the relative content of chlorophyll measured by Soil and Plant Analyzer Development.

Line 115: what do you mean with “the same imaging room” ? unclear

Response: 

Yes, Kinect sensor and SOC710 hyperspectral imager were in the same imaging room.

Line 151: do you mean measurement range? Or do you mean the distance to the plant?

Response: 

Yes, the Kinect senor detection range is from 0.5 m to 4.5 m.

“the imaging distance was between 0.80 m and 4.00 m” has been modified to “the detection range was from 0.50 m to 4.50 m”.

Kinect sensor: Microsoft Kinect 2.0.

Figure 1: description of a) is doesn’t fit to the image

Response: 

The title of Figure 1 of the original manuscript was divided into two lines, so it was easy to misunderstand. Now, the title has been revised. 

All the pictures in the manuscript have been revised.

I don’t get why we need the Furrier Transform. Can you try to clarify this point?

Response: 

The reasons for using Fourier transform have been supplemented in 2.3.1.

It is very difficult to obtain registration parameters by using an image feature point method (such as SIFT [36], SURF [37], or Harris corner feature [38]) because (i) the image resolution of SOC710 is 696 × 510 px, and the plant image area resolution of the Kinect sensor is 170 × 170 px, the two imaging regions have large zoom multiples, (ii) the imaging region of the two sensors are inconsistent, and (iii) the plant morphology is complex.

For the image of plant area, there are mainly problems of displacement, rotation angle and zoom factor. We used SIFT, SURF, Harris corner feature and other registration methods to analyze the spectral reflectance registration methods.

As a result, SIFT, SURF and Harris registration methods can not match feature points or mismatched feature points. Spectral reflectance cannot be mapped to depth coordinates accurately, mainly these methods are based on image feature points to achieve registration. But the phase correlation method, image Fourier transform is used to calculate the registration parameters in frequency domain, the registration accuracy is better. The registration performance of 60 tomato plants is stable.

There is no result of other registration methods added in this paper for the time being, mainly because most of the other registration methods fail to register, so it is meaningless to count their registration evaluation indexes (such as normalized gray scale similarity coefficient, spectral overlap rate and mutual information value).

How are the stickers detected? Manual selection? Algorithm? Color?

Response: 

In this study, automatic recognition label sticker by color threshold was used, relevant instructions have been marked in red font in manuscript 2.3.2.

During the whole plant growth cycle, the canopy morphology changes greatly. In order to acquire suitable plant images, it is necessary to adjust the relative position of sensors and plants. Therefore, the measurement system needs self-calibration function in order to meet the needs of periodic measurement. We need to solve the problem of autonomous registration of multiview point cloud models.

In a fixed imaging room environment, it is easy to identify the label area, which lays the foundation for fast rough registration of multiview point clouds. Only when rough registration is completed, ICP precise registration method can be used. Otherwise, point cloud models from different perspectives can not achieve accurate registration.

2.4. Data processing:

How did the ICP perform with 90° movement interval? I would suggest it failed in most cases to align as the ICP need at least 80% overlapping points to work. Did you use fixed angles instead?

Response: 

In the previous study, we explored the effect of the perspectives number on reconstruction accuracy. The rigid object (standard ball) and non-rigid object (tomato plant) were selected. Each plant RGB-D images were collected from 12 perspectives, the interval is 30 degrees. The recognition accuracy of stickers, the effect of perspectives number on reconstruction accuracy, and the effect of perspectives number on morphometry were analyzed systematically when the reconstruction angle intervals were 60, 90 and 120 degrees, respectively.

The results show that the point cloud model registration by rough registration firstly, then ICP precise registration. The point cloud model registration can be achieved by 60, 90, and 120 degrees, respectively. However, if ICP registration was used directly, can not achieve multiview point cloud registration.

Plants are non-entity measurement objects, and each view angle is partially occluded. When the angle interval is 90 degrees, the reconstruction accuracy requirement can be met.

This study focuses on plant multispectral three-dimensional reconstruction. Due to the space limitation of the paper, the effect of the number of views on reconstruction accuracy is not given in this paper.

Did you perform ICP matching with more angles? Could you provide a 3D image of the resulting matched point cloud? Did you now just used 4 AOVs per plant or more views for the reconstruction?

Response: 

In the previous study, each plant RGB-D images were collected from 12 perspectives, the interval is 30 degrees, plant point cloud models were reconstructed by three, four and six perspectives RGB-D images respectively. The error between reconstructed point cloud and scanned point cloud was evaluated by Hausdorff distance set. The results showed that four perspectives could satisfy the precision requirement of plant three-dimensional point cloud model reconstruction (It can be observed whether the point cloud model of cultivation pot is complete or not). Increasing the number of views will affect the efficiency of plant reconstruction and measurement.

In this study, using fixed angle interval will simplify the registration process, achieve fast rough registration of multiview point cloud, and improve the success rate of ICP registration.

This study focuses on plant multispectral three-dimensional reconstruction. Due to the space limitation of the paper, the effect of the number of views on reconstruction accuracy is not given in this paper.

Line 308: you don’t have to name every time all indexes you use. Please change this in the paper.

Response: 

Yes, we have revised the relevant description in the manuscript.

Figure 6 is useless please delete, or give result as pillow diagram of all plants as a mean value with error bars

Response: 

Figure 6 has been modified. Figure 6a and 6b of the manuscript were placed in an image, showing only the average values of each band and the standard deviation of the mean spectral reflectance from different perspectives, and the relevant descriptions in the manuscript has been revised.

How did you crop out the pot from the point cloud?

Response: 

Because the center coordinate of rotation axis was defined as (0, 0, 0), when the plant point cloud model is reconstructed, all points were translated (as shown in Form 7). The plant cultivation pot was placed on the electric rotary table. The height of the cultivation pot is known, so the plant canopy point cloud model above the pot can be selected according to the height.

Figure 7: Z axis looks weird. 3D view possible? Could you show the registration result? Or is this just one shot?

Response: 

Figure 7 is a three-dimensional view. Because it shows only the canopy of the plant, it looks weird. The results of plant registration in different bands are given in Figure 5m-5q of the manuscript.

The results of tomato plants multispectral three-dimensional point cloud models are all available. Due to the space limitation of the paper, only one tomato plant treatment process diagram is represented.

How was the outcome of the different nutrition to the chlorophyll content of the tomato plants? Which tomatoes had which real value (ground truth)?

Response: 

Chlorophyll content is an important parameter for judging the normal growth of cucumber plants. There is a significant correlation between chlorophyll content and nitrogen content. The regulation of plant nitrogen fertilizer can be realized by measuring SPAD value of plant canopy. (Nitrogen is a major nutrient element in plants, which is an important condition for healthy growth of plants)

Conclusion is too long and mixed with discussion and results. Please shorten it to the necessary outcomes and the “golden nugget” of the paper

Response: 

The conclusion has been revised and marked in red font.

The manuscript has been revised according to the expert's opinions, and answered the relevant questions. The whole text has been checked and revised uniformly. The language of the revised manuscript has been re-edit by MDPI English Editing and AJE (American Journal Experts). Please review it again. Thank you very much.

Yours sincerely,

Guoxiang Sun, Ph.D.

College of Engineering, Nanjing Agricultural University, China

Reviewer 2 Report

Summary:

This paper presents a method to reconstruct a full 3D model of a plant with multispectral information. The authors propose a system where the plant is set on a turn table that rotates in front of a Kinect and a multispectral sensor. Multiple measurements taken at different rotation angles are merged to generate the full 3D model of the plant. The system is pre-calibrated in order to transfer the multispectral information to the attributes of the 3D point cloud. A concrete application of the system is given with the analysis of the chlorophyll content of tomato plants.

Strengths:

The paper is in general well written and easy to read.

The authors propose a nice application of 3D reconstruction of plants with adding multispectral information. The authors also present a complete system that is able to reconstruct accurate 3D models augmented with multispectral information.

I found interesting to apply the Fourier based registration method to Kinect and multispectral images.

Weaknesses:

My strongest critic is that the technological novelty is limited because the proposed system is a combination of existing techniques. In addition the problem of reconstructing 3D models with a turning table and adding the multispectral information after careful calibration seems not challenging and already solved in other applications.

I found not clear the motivation why to use the Fourier-based registration method for calibrating the Kinect and multispectral sensors. With using a checkerboard, a simple state-of-the-art calibration method would work fine, isn’t it?

Minor comments about the writing:

In the abstract, giving details of the results is not necessary (l.25-26, l.30-31). Discussion in a more abstract level would be better.

In general figures should be revised. Legends such as in figure 1. are difficult to understand, sub captions like (a), (b) … are sometimes misplaced (fig. 4); font is too small like in figure 2. Figure 8 is cut in two pages. Figure 6 should be split because it is too difficult to read in the current form.

In section 3.2. l 400-422. The discussion is a bit obvious: that there are more points in the full 3D model and thus more information to do analysis. I think this part can be shortened.

l.352: “thpot“ -> “the pot”

Rating:

Also the technical contribution is not so original, the authors propose a nice system with an interesting application and analysis of tomato plants. If the authors correct the small problems in the figures and text (explaining difficulties and also reasons why to use the Fourier-based registration method instead of the standard key-points based calibration method), I think this would be an interesting paper.

Author Response

Response to Reviewer 2 Comments

Dear Reviewers and Editor,

We have revised the manuscript based on the Reviewers’ comments, and each change made in response to these comments is marked in red font in the revised manuscript (attached). Below, please find our responses to Reviewers’ comments:

Comments and Suggestions for Authors

Summary:

This paper presents a method to reconstruct a full 3D model of a plant with multispectral information. The authors propose a system where the plant is set on a turn table that rotates in front of a Kinect and a multispectral sensor. Multiple measurements taken at different rotation angles are merged to generate the full 3D model of the plant. The system is pre-calibrated in order to transfer the multispectral information to the attributes of the 3D point cloud. A concrete application of the system is given with the analysis of the chlorophyll content of tomato plants.

Strengths:

The paper is in general well written and easy to read.

The authors propose a nice application of 3D reconstruction of plants with adding multispectral information. The authors also present a complete system that is able to reconstruct accurate 3D models augmented with multispectral information.

I found interesting to apply the Fourier based registration method to Kinect and multispectral images.

Weaknesses:

My strongest critic is that the technological novelty is limited because the proposed system is a combination of existing techniques. In addition the problem of reconstructing 3D models with a turning table and adding the multispectral information after careful calibration seems not challenging and already solved in other applications.

Response: 

Yes, three-dimensional reconstruction technology has become increasingly mature, and there are many reconstruction methods. At the same time, there are many studies on the detection of plant nutrients by multispectral or hyperspectral imaging technology (mainly based on two-dimensional images). However, most of the nondestructive detection technology, morphological detection and physiological information detection for plant are two independent measurement systems.

In order to solve the problem of repeated calibration for periodic measurement of plant information, an autonomous calibration method was proposed, to solve the multiview point clouds registration and spectral reflectance registration, achieving multispectral three-dimensional point cloud model reconstruction.

It lays a foundation for plant morphology measurement and physiological information diagnosis, and forms a highly integrated measurement system (A system completes the measurement of three-dimensional geometric and physiological information). It provides a better measurement scheme for plant phenotype detection.

I found not clear the motivation why to use the Fourier-based registration method for calibrating the Kinect and multispectral sensors. With using a checkerboard, a simple state-of-the-art calibration method would work fine, isn’t it?

Response: 

In the process of research, we used chessboard and plant images respectively, which can accurately calculate the transformation parameters.

Now the chessboard registration map has been changed to plant registration image.

The reasons for using Fourier transform have been supplemented in 2.3.1.

It is very difficult to obtain registration parameters by using an image feature point method (such as SIFT [36], SURF [37], or Harris corner feature [38]) because (i) the image resolution of SOC710 is 696 × 510 px, and the plant image area resolution of the Kinect sensor is 170 × 170 px, the two imaging regions have large zoom multiples, (ii) the imaging region of the two sensors are inconsistent, and (iii) the plant morphology is complex.

For the image of plant area, there are mainly problems of displacement, rotation angle and zoom factor. We used SIFT, SURF, Harris corner feature and other registration methods to analyze the spectral reflectance registration methods.

As a result, SIFT, SURF and Harris registration methods can not match feature points or mismatched feature points. Spectral reflectance cannot be mapped to depth coordinates accurately, mainly these methods are based on image feature points to achieve registration. But the phase correlation method, image Fourier transform is used to calculate the registration parameters in frequency domain, the registration accuracy is better. The registration performance of 60 tomato plants is stable.

There is no result of other registration methods added in this paper for the time being, mainly because most of the other registration methods fail to register, so it is meaningless to count their registration evaluation indexes (such as normalized gray scale similarity coefficient, spectral overlap rate and mutual information value).

Minor comments about the writing:

In the abstract, giving details of the results is not necessary (l.25-26, l.30-31). Discussion in a more abstract level would be better.

Response: 

The abstract and conclusion have been revised and marked in red font.

In general figures should be revised. Legends such as in figure 1. are difficult to understand, sub captions like (a), (b) … are sometimes misplaced (fig. 4); font is too small like in figure 2. Figure 8 is cut in two pages. Figure 6 should be split because it is too difficult to read in the current form.

Response: 

All the pictures in the manuscript have been revised.

In section 3.2. l 400-422. The discussion is a bit obvious: that there are more points in the full 3D model and thus more information to do analysis. I think this part can be shortened.

Response: 

Yes, we have shortened this section.

l.352: “thpot“ -> “the pot”

Response: 

Yes, the description in the manuscript is incorrect.

thpot” has been modified to “the pot” .   

Rating:

Also the technical contribution is not so original, the authors propose a nice system with an interesting application and analysis of tomato plants. If the authors correct the small problems in the figures and text (explaining difficulties and also reasons why to use the Fourier-based registration method instead of the standard key-points based calibration method), I think this would be an interesting paper.

Response: 

We have revised it in accordance with the expert reviewer's opinion.

The manuscript has been revised according to the expert's opinions, and answered the relevant questions. The whole text has been checked and revised uniformly. The language of the revised manuscript has been re-edit by MDPI English Editing and AJE (American Journal Experts). Please review it again. Thank you very much.

Yours sincerely,

Guoxiang Sun, Ph.D.

College of Engineering, Nanjing Agricultural University, China

Round 2

Reviewer 1 Report

Quality of the paper improved a lot. Just some minor spelling errors and errors in the citations should be corrected before publication.